# The Effects of Frost and Fire on the Traits, Resources, and Floral Visitors of a Cerrado Plant, and Their Impact on the Plant–Visitor Interaction Network and Fruit Formation

**DOI:** 10.3390/plants14131977

**Published:** 2025-06-28

**Authors:** Gabriela Fraga Porto, José Henrique Pezzonia, Ludimila Juliele Carvalho Leite, Jordanny Luiza Sousa Santos, Kleber Del-Claro

**Affiliations:** 1Programa de Pós-Graduação em Entomologia, Faculdade de Filosofia, Ciências e Letras de Ribeirão Preto—FFCLRP, Universidade de São Paulo—USP, Ribeirão Preto 14040-901, Brazil; gabrielafraga@usp.br (G.F.P.); jh.pezzonia@usp.br (J.H.P.); ludimila.leite@usp.br (L.J.C.L.); 2Instituto de Biologia, Universidade Federal de Uberlândia, Uberlândia 38400-902, Brazil; jordanny.santos@ufu.br

**Keywords:** mutualism, pollination, tropical savanna, plant–insect interactions

## Abstract

The Cerrado, the world’s most diverse savanna, has several adaptations to fire. However, intense and frequent fires, especially after frosts, can severely impact this ecosystem. Despite this, few studies have evaluated the combined effects of frost followed by fire. We investigated how these disturbances affect plant traits, floral resources, floral visitor richness, and the structures of plant–pollinator interaction networks by using *Byrsonima intermedia*, a common Malpighiaceae shrub, as a model. We compared areas affected by frost alone and frost followed by fire and the same fire-affected area two years later. We examined pollen, oil volume, buds, and racemes and recorded floral visitors. Our main hypothesis was that fire-affected areas would exhibit higher floral visitor richness, more conspicuous plant traits, and greater fruit production than areas affected by frost only, which would show higher interaction generalization due to stronger negative impacts. The results confirmed that frost drastically reduced floral traits, visitor richness, and reproductive success. In contrast, fire facilitated faster recovery, triggering increased floral resource quantities, richer pollinator communities, more specialized interactions, and greater fruit production. Our findings highlight that fire, despite its impact, promotes faster ecosystem recovery compared to frost, reinforcing its ecological role in the Cerrado’s resilience.

## 1. Introduction

The Cerrado is an ecosystem strongly shaped by fire, which has historically determined its structure, composition, and functioning [1,2]. The species that comprise its biota have developed a set of adaptations that favor persistence in environments subject to recurrent fires. These adaptations include structural traits such as thick bark, protected buds, and underground reserve organs as well as life-cycle strategies like high resprouting capacity and synchronization of activity periods with the post-fire increase in resource availability [3,4,5].

Although fire plays a central ecological role in the Cerrado, extreme climatic events such as frost can also significantly impact this region’s biodiversity. Despite occurring sporadically, frost events cause high mortality for aboveground biomass, primarily affecting young individuals and cold-sensitive species [6,7]. When these events occur sequentially or coincide temporally, their effects may be intensified, leading to changes in vegetation regeneration, species functional traits, and the dynamics of ecological interactions [8].

Mutualistic interactions between plants and pollinators are fundamental to maintaining biodiversity, as they ensure plant reproduction and promote gene flow [9,10,11]. The formation and stability of these interactions depend on floral traits such as flower abundance, phenology, and the quantity of floral resources offered, in addition to environmental conditions like temperature and precipitation [12,13,14,15]. Changes in these factors can affect pollinator behavior, interaction frequency, and the organization of ecological networks [16,17,18,19].

In the Cerrado, there is evidence that fire directly influences flowering patterns, the availability of floral resources, and the composition of flower visitor communities [20,21]. These effects often favor generalist species, which are more flexible in terms of partner selection and therefore more resilient [22]. In contrast, specialist species tend to be more sensitive to environmental changes [23,24]. Consequently, ecological networks in fire-affected areas usually exhibit lower degrees of specialization, which, despite increasing the cohesion of interactions in the short term, may reduce stability in the face of new disturbances [25,26,27,28].

Despite our extensive knowledge of the effects of fire, little is known about how frost, either alone or in combination with fire, impacts vegetative traits, floral resource availability, and plant–pollinator interaction networks in the Cerrado. This knowledge gap becomes particularly relevant given the increasing frequency of extreme climatic events associated with global climate change [7,8,29,30].

In this context, we investigated the effects of frost and frost followed by fire on vegetative traits, floral resources, visitor richness and abundance, and the structure of interaction networks of *Byrsonima intermedia* A.Juss. (Figure 1), a key species in the Cerrado. We evaluated these parameters shortly after a severe frost event that affected a Cerrado reserve in July 2021. Two months later, the same area was impacted by a large wildfire that lasted three days and burned almost the entire reserve, leaving only a small area, less than 10%, unaffected by fire. Additionally, we reassessed plant traits, resources, and plant–pollinator interaction network structure and fruit formation two years after frost and fire to understand how these events influence interactions over time. This unique sequence of events provided an opportunity to simultaneously investigate the effects of frost and frost followed by fire on plant–floral visitor interaction networks.

Based on the high resilience of fire-adapted species [31,32], we hypothesized that the area affected by frost followed by fire would exhibit higher visitor abundance and richness, greater development of vegetative traits, higher availability of floral resources, and greater fruit production compared to the area affected only by frost. This expectation was based on the fact that fire, in addition to removing the dead biomass left by frost, promotes canopy opening, improves light conditions, and stimulates physiological processes such as resprouting and flowering, thus accelerating vegetation recovery [8,33]. In contrast, in the area affected exclusively by frost, we expected a more generalized interaction network due to the lower availability of floral resources and reduced visitor abundance and diversity, a pattern already observed in networks affected by environmental disturbances [28,34,35].

## 2. Materials and Methods

### 2.1. Study Site

The study was conducted in a Cerrado (Brazilian tropical savanna) area within the legal reserve area of the “Clube de Caça e Pesca Itororó (CCPIU)” in Uberlândia, State of Minas Gerais, Southeastern Brazil (18°59′569″ S–48°18′351″ W). According to the Köppen classification, the climate in the region is of the AW type, with two well-defined seasons: a rainy season occurring from October to April and a dry season from May to September [36]. There are different types of vegetation in the reserve, such as open grassland or savanna with sparse or no trees (Campo limpo); grassland with scattered shrubs and small trees (Campo sujo); a savanna characterized by a combination of grasslands, scrublands, and areas with scattered trees and shrubs (Cerrado); a dense, more wooded or forested area (Cerradão); and “Veredas”, that is, palm swamps or wetland surrounded by Cerrado vegetation [37]. We operated in an area of the Cerrado that suffered heavy frost in July 2021, an area of the Cerrado that suffered frost and was burned in early September 2021, and an area where *B. intermedia* flowering occurred in 2023.

### 2.2. Study System

The plant species chosen as a model for this study was *Byrsonima intermedia* A. Juss., commonly known as “murici-pequeno” (Figure 1). The choice of this species was motivated by its abundant occurrence in the reserve and its rapid recovery after fire. This shrub can reach heights of approximately 0.5 to 2.5 meters [38]. The floral morphology of this species is similar to that of Malpighiaceae in general: five sepals with oil-producing glands (elaiophores), ten stamens, and three carpels [39]. Its anthesis is diurnal, and the flowers last an average of 48 hours [40]. This species is self-incompatible and primarily pollinated by bees, offering floral resources such as oil and pollen [41,42].

### 2.3. Measuring Richness, Abundance, and Frequency of Floral Visitors

In order to assess whether disturbances affected the richness, abundance, and frequency of floral visitors and the structure of the plant–floral visitor interaction network of *B. intermedia* in 2021, we selected 30 shrubs in the area affected only by frost and in the area affected by frost and subsequent fire. The area affected only by frost was located approximately 500 m away from the area that experienced impacts from both frost and subsequent fire. The individuals were of similar sizes (~1 m to 1.5 m in height) and separated by at least 10 m. Later, in November 2023, two years after the disturbances, we again, and in the same manner, selected thirty individuals in the area affected by frost and subsequent fire to evaluate the effect of these disturbances on traits, resources, and the plant–pollinator network structure over time. In both years of study, the plants were inspected three times a week on sunny days during the flowering period. We made direct observations lasting 40 min for each plant, between 8 am and 5 pm, with a 10 min rest interval at the end of each series. The visitors were collected and taken for identification at the Laboratory of Behavioral Ecology and Interactions (LECI) at the Federal University of Uberlândia. All collected visitors were identified using taxonomic keys [43,44] and collections previously checked by taxonomists specialized in Brazilian bees.

### 2.4. Measuring Traits and Resources of B. Intermedia

To evaluate whether the disturbances affected the plants’ resource production, we previously bagged flower buds at the pre-anthesis stage to quantify the number of pollen grains and the volume of oil produced. To quantify the volume of floral oil, we used the 30 specimens of *B. intermedia* selected for the study in each area affected by frost (2021) and frost followed by fire (2021) and the area two years post-frost followed by fire (2023). However, to quantify the number of pollen grains, we randomly selected five individuals from each disturbed area (with one bud per individual). Subsequently, the buds were taken to the Laboratory of Behavioral Ecology and Interactions, and the pollen grains were carefully removed from the anthers using tweezers and placed onto a slide to allow for counting. We used acetic carmine as a stain, as indicated by Kearns and Inouye (1993) [45], and counting was performed using a manual counter under an Olympus CX40 optical microscope. To quantify the oil volume, we used microcapillaries with a capacity of 1 µL.

To evaluate whether the disturbances affected plant growth, we counted the total number of racemes and healthy buds produced by *B. intermedia* specimens weekly near the flowering period and considered the highest number of buds before flowering to be the total number of buds produced by each individual. We preferred to count buds instead of flowers because the buds of this plant are severely attacked by parasitic wasps.

### 2.5. Measuring Formed Fruits

To investigate whether fruit production was affected by disturbances both post-frost (2021) and post-frost and post-fire (2021) and two years post-frost and post-fire (2023), we randomly marked an inflorescence on each *B. intermedia* specimen (N = 30 per disturbance area) where observations of floral visitors were made and carried out hand pollination experiments. In treatment (1), a spontaneous self-pollination test was carried out, used to measure autogamy and the need for pollinators; and in treatment (2), an artificial self-pollination test was performed to determine self-compatibility. Finally, we also carried out a control experiment in which the flowers were not manipulated and remained free for natural pollination. In the first two treatments, the flower buds were bagged with voile fabric bags to avoid visitation by pollinating agents while allowing the entry of sunlight and the exchange of gases.

### 2.6. Plant–Pollinator Interaction Network

To investigate how disturbances affect the structure of plant–pollinator networks, making them more or less specialized, three quantitative interaction matrices were constructed for each area affected by frost (2021) and frost and fire (2021) and for the areas two years post-frost and post-fire (2023). A network-level metric (H2′) and two individual-level metrics were calculated using the ‘bipartite’ package in R, following recommendations on weighted interaction metric analysis from the package documentation [46,47] (Dormann et al., 2008, 2009). The network-level metrics were calculated using the ‘networklevel’ function, while the species-level indices were quantified using the ‘specieslevel’ function [46].

We measured the specialization of each area under the influence of disturbances in network level (H2′). This is a measure based on the deviation of a species’ realized number of interactions from what is expected based on the total number of interactions. The more selective the species, the greater the value of H2′ for the web, with 0 indicating no specialization and 1 indicating complete specialization [48,49]. To assess if an observed network structure is a result of randomness, 1000 null matrices were constructed using the Patefield method (the ‘r2dtable’ method in the nullmodel function of the bipartite package). The Patefield method generates relatively unconstrained null models controlling only for the network dimensions and the condition that marginal totals remain identical to the observed network [47,50].

The species-level metrics calculated were specialization (d’), which measures the deviation of the observed interaction frequencies of the focal species from an expected interaction frequency of a perfectly generalized species (with interactions evenly distributed among all partners), for which values range from 0 (extreme generalization) to 1 (specialization) [48], and degree, which is the number of links per species, i.e., the number of plant species with which each pollinator species interacts [51].

### 2.7. Data Analysis

To investigate whether the richness and abundance of floral visitors, the number of floral racemes and buds, the volume of oil, and the number of pollen grains differed between the areas under the influence of the different disturbances, we used Generalized Linear Models with negative Gaussian, Poisson, or binomial distributions (whenever superdispersion occurred). The response variables of this study were the richness and abundance of floral visitors and the traits and resources of a plant, and the explanatory variables were the disturbances (frost, frost and fire, and conditions two years after frost and fire). However, to evaluate whether the disturbances affected the fruit production of *B. intermedia* in each pollination treatment, we used a Generalized Linear Model (GLM) with a binomial distribution since the data related to incidences (absence/presence of fruits). The response variable was the number of fruits grown, and the explanatory variables were the disturbances. Finally, to compare the means of the metrics of individual networks (Specialization d’ and Degreee), we used an analysis-of-variance model (ANOVA). Assumptions of normality and homoscedasticity were examined using the “DHARMa” package [52], followed by post hoc tests with estimated marginal means using the “emmeans” package [53].

## 3. Results

### 3.1. Richness and Abundance of Floral Visitors

The richness of floral visitors varied significantly among disturbances (GLM: χ^2^ = 6.12, *p* = 0.04). The lowest richness occurred in the frost-only area (2021) (0.06 ± 0.18), and the highest occurred two years after frost and fire (2023) (0.47 ± 0.14) (Figure 2a). In contrast, visitor abundance did not differ significantly (GLM: χ^2^ = 3.77, *p* = 0.152), although there was an increasing trend over time. The lowest abundance was recorded after frost (2021) (0.31 ± 0.23), followed by frost and fire (2021) (0.80 ± 0.21), and the highest was recorded two years after frost and fire (2023) (0.90 ± 0.21) (Figure 2b).

### 3.2. Frequency of Floral Visitors

The composition of dominant floral visitors shifted across disturbances. After frost and fire (2021), *Trigona spinipes* (40%) and *Tetragonisca angustula* (28%) were most frequent. In the frost-only area (2021), *Tetragonisca angustula* dominated (45%), followed by *Paratrigona lineata* (17%). Two years later (2023), visitor composition diversified, with *Paratrigona lineata* and *Centris aenea* (17% each) and *Epicharis bicolor* (14%) being the most frequent. Several new species appeared in 2023, including *Augochloropsis* sp., *Frieseomelitta flavicornis*, *Apis mellifera*, and *Pseudaugochlora graminea* (Figure 3).

### 3.3. Traits and Resources of B. Intermedia

The vegetative traits and floral resources of *B. intermedia* differed among disturbances. The quantities of racemes (GLM: χ^2^ = 6.30, *p* = 0.036) and buds (GLM: χ^2^ = 8.51, *p* = 0.014) were lowest after frost (2021) and progressively increased after frost and fire (2021), increasing even further two years later (2023) (Figure 2c,e).

Floral oil volume also differed (GLM: χ^2^ = 9.08, *p* = 0.01), with the lowest value occurring after frost (2021) (0.07 ± 0.01) and the highest observed two years after frost and fire (2023) (0.15 ± 0.02) (Figure 2d). Pollen production exhibited the clearest pattern (GLM: χ^2^ = 135.63, *p* < 0.001), nearly doubling from 16,286 ± 946 in the frost-only area to 30,701 ± 946 two years after frost and fire (2023) (Figure 2f).

### 3.4. Fruit Production

Pollination tests confirmed that *B. intermedia* requires cross-pollination for fruit setting, as neither spontaneous nor manual self-pollination resulted in significant fruit development (GLM: χ^2^ = 0.22, *p* = 0.90; χ^2^ = 4.49, *p* = 0.11). The few fruits produced were likely a product of contamination during manipulation.

Under natural pollination conditions, fruit production differed significantly (GLM: χ^2^ = 9.86, *p* = 0.007) (Figure 4). The highest fruit-setting level was observed two years after frost and fire (2023) (3.3 ± 1.0), and this figure was followed by those for the frost-only area (2021) (1.0 ± 0.41) and the area affected by frost and fire (2021) (0.8 ± 0.3).

### 3.5. Plant–Visitor Interaction Network

The values of the specialization/generalization indices (H2′) at the network level were significantly higher compared to the null expectation under random association (*p* < 0.05) for the area affected by frost and fire (2021) (0.47; *p* = 0.002) and in the same area two years after frost and fire (2023) (0.42; *p* = 0.003). In the area affected only by frost (2021), the network was less specialized, although the observed value was not significant, suggesting that this result may be due to chance (0.24; *p* = 0.32; Figure 5).

Among the species-level network metrics, the average degree values were not significantly different among the disturbances (F = 0.81; DF = 2; *p* = 0.45). Similarly, the average specialization *d*’ values did not vary among the different disturbances (F = 2.49; DF = 2; *p* = 0.09), although a trend towards lower specialization was observed for areas affected by frost (2021) (Figure 6).

## 4. Discussion

The hypothesis that fire affects plant traits, resources, and plant–floral visitor interactions in the Cerrado less than frost was corroborated by our main results. The frost event of 2021 had a more negative impact on plant traits and resources, making the plant–floral visitor interaction network more generalized, resulting in lower reproductive success of *B. intermedia*. The floral traits and resources of the plants were less conspicuous in the frost-only area, and the richness of floral visitors was significantly lower. In addition, the plant–floral visitor ecological network of the area affected by frost only was the most generalized. This area had more frequent visits from generalist species and maintained a greater number of interactions with *B. intermedia*, which possibly contributed to the lower average number of fruits developed. On the other hand, plant traits, such as the number of racemes and buds, were higher in the areas affected by frost and fire. After the frost event and the subsequent fire, the resources in the area increased significantly two years later, and there was also a greater richness of floral visitors. The appearance of new species in the area two years after the disturbance contributed to a higher average for fruit formation.

This is the first study to investigate the effects of frost and frost followed by fire on traits, floral resources, and plant–visitor floral interactions for a Cerrado plant species. In this study, we observed that the richness of floral visitors decreased considerably after frosts, especially species considered specific pollinators, such as oil-collecting species (e.g., members of the genera *Epicharis* and *Centris*) [39,54]. Previous studies conducted in other locations also showed that low temperatures have a negative impact on pollinator richness and floral visitors [55,56,57] and lead to a deficiency in the development of pollen and ovules [58,59]. In our study, the area affected by frost had the fewest pollen grains produced per bud, especially when compared to the area evaluated two years after frost and fire. This indicates that the low temperature caused by frost possibly causes low pollen production by *B. intermedia* as well as lower quantities of floral oil, racemes, and buds. Nevertheless, the plants in the Cerrado present efficient fire adaptation, responding to this physical stimulus by resprouting and blossoming [60].

Previous studies have shown the direct effects of frost on leaves, flower buds, and plant growth [7,37,61,62]. However, few studies have been conducted in tropical savannas [63,64], and among these, none have focused on plant traits that directly affect interactions with animals. Evidence from other ecosystems suggests that frost can indeed impact floral traits relevant to pollination. For instance, Pardee et al. [65] reported a 40% decrease in the number of flowers developed by a plant species exposed to frost in Colorado, USA. Additionally, this species experienced 48% fewer pollinator visits and an 8.5% reduction in fruit setting. Similarly, in our study, the area affected by frost in the Cerrado produced the lowest average number of racemes and buds, indicating that frost can impair the growth and development of floral structures essential for attracting pollinators. This area also exhibited lower network-level specialization and a higher frequency of visits by generalist floral visitors, such as *Tetragonisca angustula* and *Paratrigona lineata*, a pattern comparable to that observed for *B. intermedia* in disturbed areas of the Cerrado [66,67].

Although generalist species are considered critical in disturbed environments, as they participate in most of the links established with plant species [23,68,69], there is limited understanding of the ecosystem services provided by generalist pollinators. Burns et al. [70] showed that the ability to attract many pollinator species (high generalization) in disturbed locations resulted in decreased seed-by-ovule ability (fitness), i.e., decreased plant reproductive capacity. Here, we found that the average number of fruits formed was lower in the area under the influence of frost alone compared to the area under the influence of frost followed by fire. This shows that frost has currently unknown indirect and direct effects on the reproductive capacity of Cerrado plants.

In contrast to the area under the influence of frost only, in the area under the influence of frost and subsequent fire, both in the first flowering period and two years after, there was a greater presence of specific pollinators as well as more conspicuous traits and resources. Studies have shown positive changes in resource availability and post-fire pollinator communities [4,71,72], which may result in changes in network specialization [73,74,75]. In this case, the specialization of the network in the area under the influence of frost followed by fire was similar in the first flowering and two years after the disturbance (0.47 and 0.42, respectively). This may be linked to the greater availability of resources and the greater attraction of specific pollinators, especially two years after the disturbances, when new species colonized the disturbed area. Despite this, we did not detect significant differences in the metrics at the species level evaluated, although the specialization of the species tended to decrease in the area under the influence of frost only, reinforcing the notion that the interactions were less specialized.

Although fire has played an important evolutionary role in the formation of the Cerrado [1,32], its high frequency, driven by anthropogenic factors, may negatively affect species and their interactions [76,77,78]. We found that two years after frost and fire, important bee species that had disappeared reemerged. This includes *Paratetrapedia*, a key genus of oil-collecting bees in the Americas [79], and *Epicharis analis* and *E. flava*, both associated with Malpighiaceae species [41]. These observations reinforce the importance of conducting long-term studies to understand the direct and indirect effects of anthropogenic disturbances and climatic extremes on ecological communities. Our results also have practical implications for fire management in the Cerrado. While high fire frequency can be detrimental [78,80], fire following frost may help restore floral traits and pollinator communities, mitigating some of the negative impacts of frost. This suggests that indiscriminate fire suppression could inadvertently hinder natural regenerative processes essential for maintaining plant–pollinator interactions in this fire-adapted ecosystem. Nevertheless, our study has limitations. The short temporal scale may not fully capture long-term dynamics, and focusing on a single plant species and few plots limits broader generalization. Expanding the temporal and spatial scales and including multiple species would be crucial to facilitate more effective conservation and management strategies under increasing climate variability.

## 5. Conclusions

Our study reveals the complex dynamics between floral traits, resources, and plant-visitor interactions in response to different environmental disturbances in the Cerrado. We observed that frost, in isolation, had a significant negative effect, reducing the conspicuity of floral traits and the richness of visitors, resulting in a more generalized ecological network and lower reproductive success of *B. intermedia*. In contrast, areas that suffered the combined effects of frost followed by fire showed a remarkable recovery of floral traits and resources, with an increase in visitor richness and specialization of interactions, which culminated in greater reproductive success. These findings corroborate the hypothesis that compared to frost, fire has a less negative impact on the traits, resources, and networks of floral visitors of Cerrado species. In addition, this study highlights the importance of conducting long-term research to understand the effects of environmental disturbances, especially those promoted by anthropogenic activities, on ecological communities. These results highlight the need for mitigation strategies aimed at preserving species and maintaining ecological interactions in the Cerrado.

## Figures and Tables

**Figure 1 plants-14-01977-f001:**
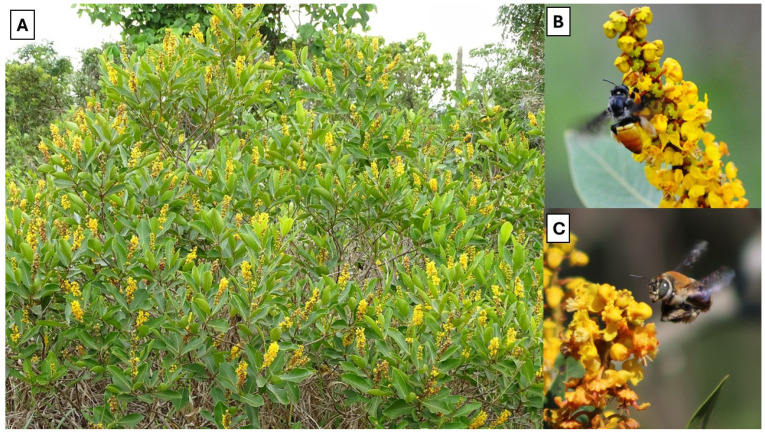
A typical shrub of *Byrsonima intermedia* (**A**), with flowers being visited by Centridini bees (Apidae): *Epicharis bicolor* (**B**) and *Centris aenea* (**C**).

**Figure 2 plants-14-01977-f002:**
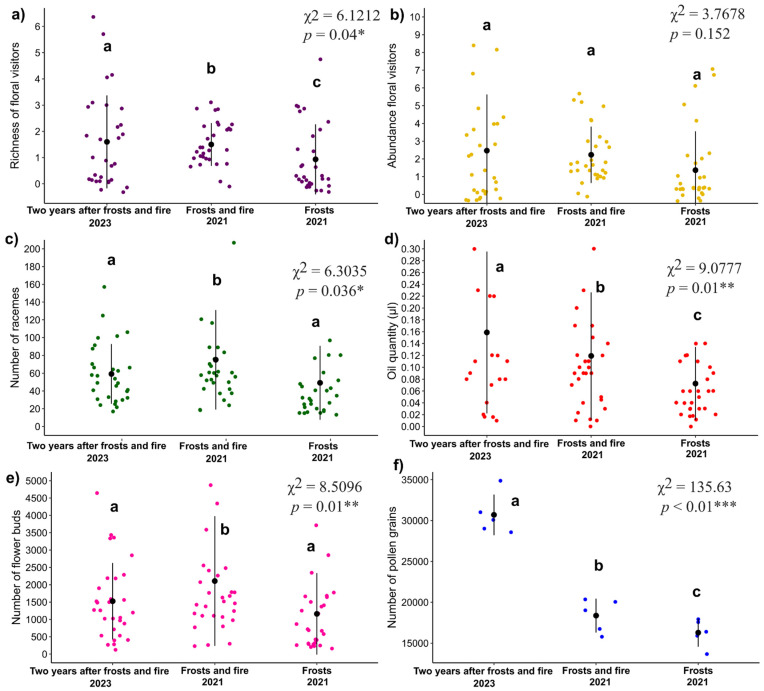
Effects of disturbances on *B. intermedia* two years after frost and fire (2023), after frost followed by fire (2021), and after frost only (2021). Variables analyzed: (**a**) floral visitor richness, (**b**) floral visitor abundance, (**c**) number of racemes, (**d**) oil volume, (**e**) number of floral buds, and (**f**) pollen quantity. Different letters indicate statistically significant differences between treatments. * *p* < 0.05; ** *p* < 0.01; *** *p*< 0.001.

**Figure 3 plants-14-01977-f003:**
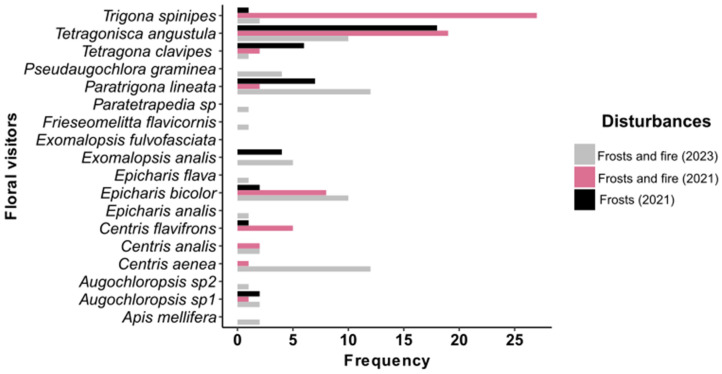
Frequency of floral visitors in *B. intermediates* two years after frost and fire (2023), under the influence of frost and fire (2021), and under the influence of frost only (2021).

**Figure 4 plants-14-01977-f004:**
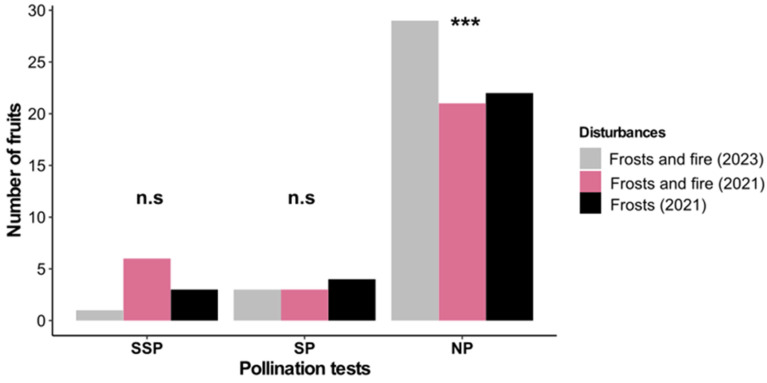
Difference in the number of fruits formed between pollination treatments (SSP—spontaneous self-pollination, SP—self-pollination, and NP—Natural pollination) two years after frost and fire (2023), under the influence of frost and fire (2021), and under the influence of frost only (2021). The symbols *** and n.s. indicate significant and non-significant differences, respectively.

**Figure 5 plants-14-01977-f005:**
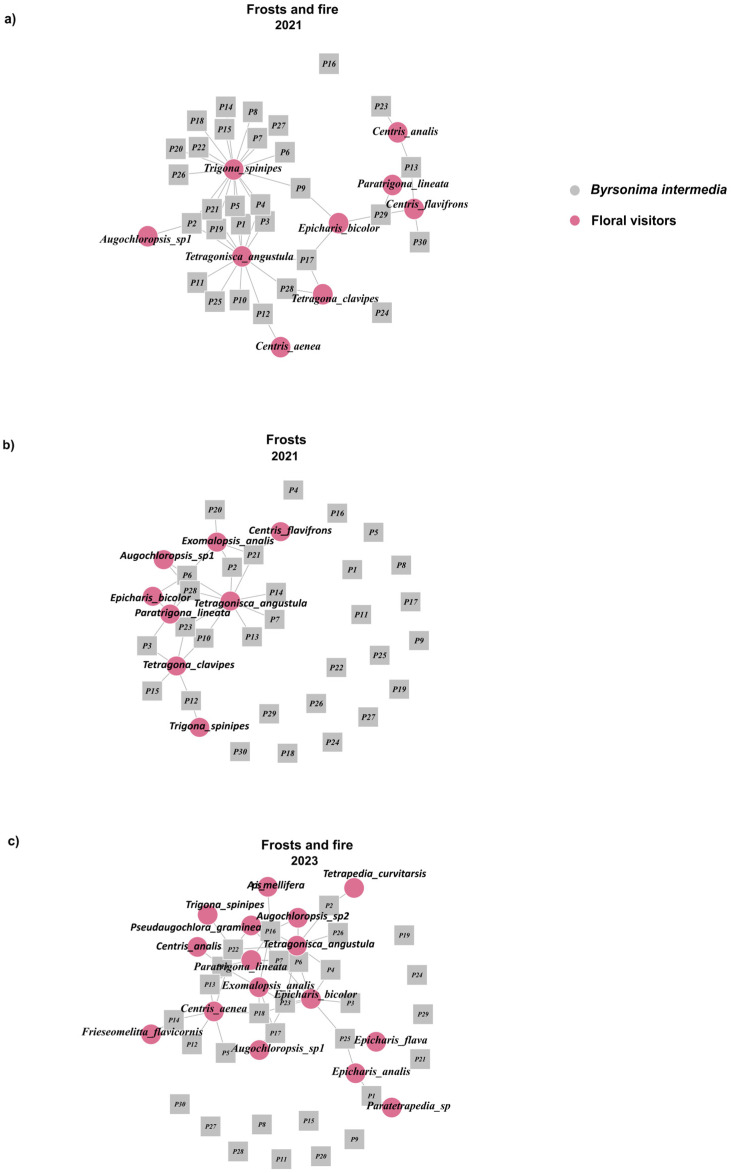
Interaction networks between individuals of *Byrsonima intermedia* and their floral visitors (**a**) after being subjected to frost only, (**b**) under the influence of frost and fire (2021), and (**c**) two years after frost and fire (2023). In the networks, the pink circles represent the floral visitors, and the gray squares signify the *B. intermedia* specimens.

**Figure 6 plants-14-01977-f006:**
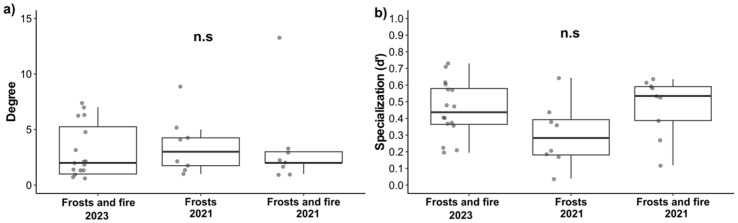
Average of the d’ specialization (**a**) and degree (**b**) values for floral visitors of *Byrsonima intermedia* two years after frost and fire (2023), under the influence of frost and fire (2021), and under the influence of frost only (2021). The ANOVA did not show significant differences (*p* > 0.05). n.s. indicates non-significant differences.

## Data Availability

The data presented in this study are available on request from the corresponding author.

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
