# Peer review of "The Effects of Frost and Fire on the Traits, Resources, and Floral Visitors of a Cerrado Plant, and Their Impact on the Plant–Visitor Interaction Network and Fruit Formation"

_plants, 2025, doi:10.3390/plants14131977_

Round 1
Reviewer 1 Report
Comments and Suggestions for Authors
Reviewer’s Comments
General Comments:
This manuscript titled “The effects of frost and fire on the traits, resources, and floral visitors of a Cerrado plant, and their impact on the plant-visitor interaction network and fruit formation” addresses the important ecological function of fire, challenging traditional assumptions that it only has negative impacts on ecosystems. The study highlights how fire can promote faster ecosystem recovery compared to frost, thereby reinforcing the ecological role of fire in maintaining the resilience of Cerrado ecosystems. This topic is interesting and has potential implications for ecosystem management, conservation, and theory development.
However, there are several points that should be addressed before the manuscript can be considered for publication.

Author Response
Major Comments: 1. Introduction: clarity and focus
Thank you very much for the valuable comments. We have revised the introduction according to your suggestions, emphasizing the role of fire in the Cerrado for both species dynamics and ecological interactions. Additionally, we improved the discussion on frost effects, making the text more focused and aligned with the study’s objectives.
Results section: brevity and clarity
We appreciate the comments and have taken into account the suggestion to make the results clearer and more concise. We believe the text is now more fluid and the data easier to understand.
Regarding Figure 5, we increased the font size of the figure legend. However, it was not possible to enlarge the figure itself because the R package used automatically rearranges the network layout when its size is changed, which would compromise visual comparison among networks. We made all possible adjustments to improve visibility and recommend zooming in on the figure for better interpretation.
Discussion: context and broader comparisons
Thank you for the insightful comments and suggestions. We have incorporated additional references as suggested, within the limits of availability, since studies on frost effects in Neotropical savannas remain scarce in the literature.

Reviewer 2 Report
Comments and Suggestions for Authors
This is a thorough examination of the impact of two environmental factors, frost and fire, on the development of Byrsonima intermedia and it's pollinators. The authors do an excellent job placing their work in context and providing background on the potential interactions associated with frost and fire. Their hypothesis is well-stated and the evidence in support of it is comprehensive and convincing. The only significant suggestion I have is that the study is based on the premise that the frost only and frost plus fire treatments differ only based on the occurrence of fire and not other factors (like topology, soil characteristics, etc.). I think it would be helpful for the authors to make this assumption explicit and to document, as best as they can, that the only treatment difference is fire. (Essentially, because there is a post hoc assignment of treatments, a critic might argue that the assumption of random treatment assignment is violated. I don't agree with this point, but I think it is wise for the authors to defend themselves from this potential criticism.) On a couple minor points, the sentence starting on line 138-140 is incomplete as written (I think it needs to be coupled to the following sentence), and on line 296 B. intermedia needs to be italicized.
Author Response

(The authors gave the same response as above.)
